# Race and 1918 Influenza Pandemic in the United States: A Review of the Literature

**DOI:** 10.3390/ijerph16142487

**Published:** 2019-07-12

**Authors:** Helene Økland, Svenn-Erik Mamelund

**Affiliations:** 1Department of Business, History and Social Sciences, University of South-Eastern Norway, Raveien 215, 3184 Borre, Norway; 2Work Research Institute, OsloMet—Oslo Metropolitan University, PO. Box 4, St. Olavs Plass, 0130 Oslo, Norway

**Keywords:** influenza, pneumonia, pandemic, inequality, race, morbidity, mortality, case fatality, 1918, USA

## Abstract

During epidemics, the poorest part of the population usually suffers the most. Alfred Crosby noted that the norm changed during the 1918 influenza pandemic in the US: The black population (which were expected to have higher influenza morbidity and mortality) had lower morbidity and mortality than the white population during the autumn of 1918. Crosby’s explanation for this was that black people were more exposed to a mild spring/summer wave of influenza earlier that same year. In this paper, we review the literature from the pandemic of 1918 to better understand the crossover in the role of race on mortality. The literature has used insurance, military, survey, and routine notification data. Results show that the black population had lower morbidity, and during September, October, and November, lower mortality but higher case fatality than the white population. The results also show that the black population had lower influenza morbidity prior to 1918. The reasons for lower morbidity among the black population both at baseline and during the herald and later waves in 1918 remain unclear. Results may imply that black people had a lower risk of developing the disease given exposure, but when they did get sick, they had a higher risk of dying.

## 1. Introduction

The 1918–1919 “Spanish Flu” Pandemic infected a third of the global population, killing an estimated 50–100 million globally (2.5%–5.0%) and 675,000 in the United States (0.7%) [1,2,3]. The only year in the 20th century when black people in the USA had lower influenza mortality than white people was 1918. One hypothesis is that black people, who mainly lived in the South and under miserable living- and working conditions, cramped conditions, white racism and violence, and poor medical care, were less susceptible to the 1918 influenza pandemic autumn wave due to higher exposure to the less virulent spring and summer waves [2,3]. However, this hypothesis, the mechanisms for the crossover in the role of race in 1918 pandemic mortality, and the subsequent return to the “normal” pattern of higher black than white mortality in 1919 have received little attention in the literature, nor has this (or other) hypothesis been theoretically or empirically substantiated. It remains unclear how race as proxy for socioeconomic status was associated with disparities in, for example, household/community-level crowding and assortative mixing, or access to water and, thus, the ability to maintain a high general level of hygiene and handwashing. It is also unclear how these factors subsequently may have been associated with differential exposure to the virus and morbidity in earlier waves and how potentially acquired immunity may have buffered against mortality in later waves. To our knowledge, in this paper, we do the first review of the published data, contextual literature, and the suggested hypotheses explaining the racial differences in 1918 pandemic influenza and pneumonia morbidity, mortality, and case fatality by age, gender, and locality in the USA, reviewing studies using various sources of data and methods. The data include subgroups such as male soldiers (selectively good health) and insurance holders (a proxy of some wealth) as well as broad and representative populations gathered from survey and routine notification data (average health and wealth). Our main findings show that the black population had lower pre-pandemic and pandemic morbidity and, during September, October, and November 1918, lower pandemic mortality but higher case fatality than the white population. The reasons for higher black than white morbidity remain unclear.

## 2. Materials and Methods

We have not done a systematic review of the literature. The reason for this choice is that most of the literature on the topic, typically published from 1918 until the early 1930s, is not indexed in databases such as MEDLINE, Embase, Cinahl, SocIndex, Scopus, or Web of Science. We have therefore reviewed both contemporary and more recent literature based on our expert knowledge in the field. Reference lists of relevant known studies were screened, and other experts in the field were consulted in order to identify other additional sources. Using this method, we identified a total of 13 quantitative studies specifically researching the association between race and 1918 influenza morbidity and mortality in the United States. One key contribution of our paper is that we review studies that use different sources of data rather than only reviewing studies that rely on one specific source of data to cast light on race as proxy for socioeconomic status leading to disparities in exposure, susceptibility, access to care, and eventually disparities in 1918 influenza pandemic outcomes (based on a framework described in [4]). In the following sections, we describe the data and the strength and weaknesses of each type of data used in the reviewed literature.

### 2.1. Military Data

In 1914, the regular army of professional soldiers consisted of 127,588 troops (7.8% or 10,000 black) and 164,292 National Guard officers and enlisted men (6.1% or 10,000 black) serving state governments [5]. The army wanted to create combat divisions for the professional army, National Guard, and national army of drafted troops. Army officials expected to fill the National Guard and the professional troops with volunteers in the summer of 1917, but the expected numbers of white volunteers did not show up. Army officials therefore realized they would have to use white draftees to fill both these volunteer-oriented organizations as well as the national army of drafted troops. For the first time in American military history, draftees therefore formed the majority of the soldiers. On the armistice day on November 11, 1918, there were a total of 3.9 million US soldiers, of whom 72% were conscripted [5]. Of the nearly 380,000 black people who served during the war, almost all were conscripted (96% or 367,710).

Racial stigma played a role in the conscription process [5,6]. During the first draft call in 1917, local boards registered 24 million men (a quarter the total US population of 100 million of both genders and all ages), and medically examined 10.6 million assumed to be suitable for service (1,078,331 black people and 9,562,515 white people), ultimately producing an army of 4 million. Of the medially examined men, 51.7% of blacks and 32.5% of whites fell in Class I, a category that immediately made them eligible for induction. This higher share of black people placed in Class I continued at the same levels in subsequent draft calls. Further, draft boards did not automatically grant deferments. Fewer black people may have applied for exemptions because of illiteracy. Out of Class I individuals from the first draft in 1917, 36% of black people were inducted into the military, compared with 24% of white people. The over-inducting of black registrants also continued at the same levels throughout the war.

Black people and the foreign-born served in numbers greater than their proportion of the overall population. While 13% of enlisted men were black and 18% were foreign-born, these groups only made up 10% and 14.5% of the total population, respectively.

There was no difference in the share of black and white soldiers going to France. Out of the 380,000 black soldiers who served during the war, 52.6% or 200,000 went to France. Among the 3,513,340 white soldiers, 51.2% or 1.8 million embarked for France [5]. Out of the 200,000 black people who went to France, approximately 38,000 or 19% were combat troops. By comparison, nearly one million or 57% of the 1.8 million white troops in France were classified as combatants [5]. Over 89% of all black troops (338,000 out of 380,000) would serve in assorted labor battalions, pioneer infantry units, salvage companies, and stevedore organizations. By comparison, approximately 56% of white troops served in noncombatant units (1,973,118 out of 3,513,340). Black people made up approximately 1/3rd of the wartime army’s laboring units and 1/30th of its combat forces. The black people in labor battalions in France loaded and unloaded ships, built roads, cleaned up camps, and buried the dead [6].

The key strength of the hospital and mortality data from the military is the high probability that all cases/hospitalizations and deaths are reported. Further, although black people more often were placed in noncombatant battalions and more often lived in tent colonies fully segregated from white barracks, both black and white soldiers were at the time assumed to live under relatively controlled conditions and under the same military nutrition, clothing, training, and discipline [7]. One major weakness of military data is that they only include males 18–45 years, while other male age groups and females are not considered [7,8,9,10,11,12,13]. Military data also include the physically and mentally fittest of males 18–45 years [8,10,14,15]. Those who were not in Class I, the classification that made them eligible for immediate induction, were categorized in four other classes. Class II and III included temporarily deferred married men and skilled workers in industry and agriculture; class IV contained married men with economic dependents and key business leaders, while those unable to meet physical and mental requirements were placed in Class V. About 30% of the medically examined were considered medically unfit, including people with tuberculosis, venereal diseases, epilepsy, goiter, underweight, and nervous exhaustion [6]. Finally, although there were 12,500 American Indians in the army and 20% of the draftees were foreign-born, including 5700 Mexican Americans, and at least 9% were not US citizens [6], only “white” or “colored” (African American) were registered in the morbidity and mortality statistics; ethnicity, citizenship, language spoken, and literacy were not recorded. As we see in the next paragraphs, race is the major or only analytical category used in the insurance data, survey data, and annual population data as well.

### 2.2. Insurance Data

These data include 12 million policyholders from the Industrial Department of the Metropolitan Life Insurance Company across the USA and Canada. One of the advantages of the published insurance data is that, unlike other data, they include both monthly pneumonia and influenza (PI) mortality data for 1918–1919 and baseline PI mortality data 1911–1917 by race and gender [16]. Represented in the data, we find all industries and occupations as well as the wage-earners’ families, covering males and females aged 1 year to 75+. We may nevertheless speculate that one needed a certain level of wealth to afford a policy if it was not covered by the employer. The data may thus include the wealthier subsets of both the white and black populations. Insurance data may therefore leave out the poorest people and possibly more black than white people. Unfortunately, we do not have access to the proportion of each race by age and gender.

### 2.3. Survey Data

Survey data exist for 18 cities and some rural areas in the United States [10,14,15,17,18,19]. All households were interviewed by trained staff. The influenza cases and deaths for a household were self-diagnosed and self-reported to the data collectors, usually by a homemaker, and were of course not lab-confirmed; in 1918, it was not known that influenza was caused by a virus, so even cases brought to hospital laboratories could not have been tested. On the other hand, pneumonia cases could be easily diagnosed from auscultation and sputum smears. The influenza cases in the surveys did not only include the severe cases brought to a doctor or a hospital (or not brought to medical officers due to poor access to health care, health information, health insurance, and poverty). However, just as would be the case even today, mild cases and asymptomatic cases were probably underreported in these surveys for both races, as was also the case in official records from civilian/military doctors and hospitals.

Two studies used survey data to report on race and pandemic outcomes in 12 of these localities [10,17]. These were the city of (1) Baltimore, (2) Charles County, (3) 4 minor towns (Cumberland, Frederick, Salisbury, Lonaconing) and 3 rural communities (Frederick, Washington, and Wicomico Counties) (all in Maryland), (4) Little Rock (Arkansas), (5) San Francisco (California), (6) San Antonio (Texas), (7) Louisville (Kentucky), (8) Spartanburg (South Carolina), (9) Des Moines (Iowa), (10) New London (Connecticut), (11) Macon, and (12) Augusta (Georgia). The total population in these localities was 1.95 million, while 148,562 were sampled (7.6% of the population), covering all ages, genders, and races. However, since some areas had very few black people, results were eventually only reported in the two studies for 7–8 of the 12 areas. The canvassed areas of each locality were selected at random to make the samples representative for the population with respect to demographic variables and the impact of the pandemic upon morbidity and mortality. One of the most important factors related to both morbidity and mortality, which is observable for both the samples and the populations, is the age distributions; the sample age distributions in two of the 12 canvassed areas, the city of (1) Baltimore and (3) towns/rural areas in Maryland, did not deviate from the age distributions in the population of the canvassed areas [14]. The overall sample mortality rate was slightly lower than the population mortality rate in Maryland. We have no reasons not to believe that the 10 other canvassed areas also had samples that were representative of their respective populations.

Survey data have other advantages beyond being representative for broad groups of the population. First, because the chief feature of pandemics is much higher morbidity than during seasonal influenza epidemics (although case fatality also may rise, as in 1918), finding statistically significant differences in *morbidity* rates by subgroups is not a problem when using sample survey data. Second, although official population records are a preferable source for documenting variation in 1918–1919 pandemic mortality rates, survey data are needed to estimate reliable morbidity and fatality rates. Third, fewer people might see a doctor in mild compared to severe disease outbreaks. Therefore, survey data may better capture the actual magnitude of mild outbreaks than routine notification data. A comparison of survey data and routine notification data for Bergen, Norway, confirms this hypothesis; although the number of persons seeing a doctor for an influenza-like illness during the mild summer wave (as measured from routine notification data) was only one third of new influenza cases as reported in the survey, there were no differences in the number of influenza cases between the two sources of data during the more lethal 1918 fall wave [20].

A potential general weakness of using survey data to analyze the epidemiology of the 1918–1919 pandemic is when small subgroups are studied, for example, *mortality* or *fatality* rates by age, sex, and race. In an analysis using similar survey data from Bergen, Norway, none of the differences in fatality and mortality rates for any subgroup were statistically significant because of a low number of sample deaths [20]. The US survey data suffer from several weaknesses. First, they generally cover cities with a low percentage of black people, and the sampled areas therefore include few black people. One exception is Charles County, Maryland, where a larger share of black people was included in the survey. USA was a highly segregated society in 1918. Whether the enumerators made an effort to secure that black neighborhoods in the canvassed areas had equal probability of being randomly selected is therefore a question of debate. Second, the US data only cover the 1918 fall wave, and not waves in the spring and summer of 1918 or waves in 1919. Third, because United States participated in the First World War (WWI), the survey (and routine notification data, see next paragraph) for young adult civilian males in the United States might have been biased because 4 million males were drafted for military service, leaving possibly more frail males behind among the civilians [10,14,15]. However, the mortality rates of both males and females in urban and rural areas of one of the canvassed areas, (1) city of Baltimore and (3) rural areas, Maryland, follow the same pattern and level of morbidity and mortality by gender as in Bergen in neutral Norway [20].

### 2.4. Routine Notification Data/Population Records

This type of data includes, for example, influenza cases reported to a doctor and death certificates issued by a doctor and used in official health and population statistics. The coverage of deaths in official statistics is generally good. However, one weakness is that most states were not part of the national births and deaths registration areas in 1918 [21]; only 19 states were part of it, and most states in the South, which had the largest concentration of black people, were not part of it (e.g., Louisiana, Alabama, Georgia, and South-Carolina). Weden [22] published a paper using routine notification data and more sophisticated statistical methods than the contemporary studies, to look into the racial differences in mortality in 1918. Jordan [23] and Garret [24] also used official records to study the role of race on mortality.

### 2.5. Materials

In cases where raw data on cases/deaths and population at risk are published in the 13 identified articles, we recalculated age- and gender-specific morbidity and mortality rates by race and locality. We generally reproduced the published results. Testing for statistical significance was not done in the published literature from 1918 or in the 1920s, so our main contribution in terms of new analysis on old data is testing whether racial differences were statistically significant. In cases where both the numerator and the denominator were available, differences in pandemic outcomes were estimated using rate differences and rate ratios. A statistically significant difference in morbidity, mortality, and case fatality by race was estimated using a 1-sided z test with α = 5%, that is, z ≥ 1.65 (when alpha is 10%, 1%, or 0.1%, then the z-values are ≥1.29, ≥2.33, and ≥3.12, respectively). If data on pandemic outcomes and population at risk were available at baseline, we also recalculated excess morbidity or mortality due to the 1918 influenza pandemic by subtracting the norm during seasonal influenza. We mainly reproduced the published results. Morbidity or incidence is defined as the percentage of new cases of either influenza (I), pneumonia (P), or both (PI) in the population. Mortality is the percentage of PI deaths in the population, while case fatality/lethality is the percentage of deaths among PI cases.

For race, we use the categories black and white (people, soldiers, insurance holders, respondents). “Race” and “ethnicity” are variables that are commonly used in population health research [25,26]. Over time, race has been poorly and variably defined and has laid the groundwork for racism in various communities. There are no diseases that are exclusive to race, but geographical and social delimitations are variables that are associated with both socially-defined racial groups and disease. The concept of race, in our case, has therefore value as a social variable. We use race as a proxy for socioeconomic status in relation to morbidity, mortality, and case fatality. Much of the black population in 1918 typically lived under worse social, economic, and medical conditions than most of the white population. We thus assume that black people had lower socioeconomic status than white people did.

## 3. Results

### 3.1. Military Populations

We found one report documenting the baseline morbidity from influenza among 558,668 soldiers for the calendar year of 1917, including 545,518 white (97.6%) and 13,150 black (2.4%) soldiers. The hospital admission rate for influenza was 2.1 and 4.0 times higher for the white than for the black soldiers in 1917 and 1910–1917, respectively [7] (for 1917: 5.84% vs. 2.78%, a rate difference at 3.06, 95% CI 1.33–4.79, z = 3.47). On the other hand, hospitalization rates for lobar pneumonia in 1917, which could be due to influenza, measles, or other underlying diseases, was 4.68 times more frequent for black than for white soldiers (5.37% vs. 1.15%, a rate difference of 4.22, 95% CI 2.49–5.95, z = 4.78). There were also 2.9 times more hospitalizations for bronchopneumonia in black than white soldiers, but this difference was not significant (0.73% vs. 0.25%, a rate difference of 0.48, 95% CI 1.25–2.21, z = 0.54).

We also identified six quantitative studies researching 1918 pandemic outcomes by race in the US military. First, Howard and Love [8] investigated all soldiers in US training camps and American Expeditionary Forces (AEF) soldiers stationed in France for the calendar year of 1918. There were an estimated 4,412,533 million US soldiers in the fall of 1918 [5], of whom approximately one half was in the US and the other half in France, but morbidity and mortality data in this study were only included for 2,309,922 soldiers, of whom 2,142,336 (92.7%) were white soldiers and 167,586 (7.3%) were black soldiers. They found that morbidity, mortality, and lethality from influenza with or without complications (broncho- and lobar pneumonia, and others) were lower for white than for black troops (morbidity 26.6% vs. 29.8%, z = −12.53; mortality 0.9% vs. 1.3%, z = −1.54 (ns.); lethality 0.3% vs. 0.4%, z = −2.05). However, morbidity for troops stationed in the USA was a little higher among white than black soldiers (36.1% vs. 35.4%), but mortality was still lower for white vs. black troops (1.2% and 1.5%) (here we do not have the data to check for statistical significance). In France, the soldiers’ morbidity was lower than for those stationed at home, but mortality was higher. Morbidity in France was lower for white than for black troops (14.4% vs. 20.6%), while mortality was still lower for white vs. black troops (5.2% and 8.6%) (here we do not have the data to check for statistical significance).

Howard and Love [8] also studied morbidity from influenza and *primary* pneumonia combined by race and state for troops in US and Europe combined. The ratio of morbidity for white to that of black soldiers was 100 or over for 41 of the states, meaning higher morbidity for the white vs. black soldiers, and the ratio ranged from 108 (New Jersey) to 288 (South Carolina). In nine states, this ratio was below 100, meaning lower morbidity for white soldiers vs. black soldiers in Washington (98), the District of Columbia (97), California (93), Nevada (81), Idaho (75), Arizona (71), Wyoming (58), Montana (48), and Alaska (46). Although a substantial number of black people moved from the South to the North as part of the Great Migration starting in 1916, the morbidity rates for black soldiers in other states than those to the South are unreliable. However, if we consider the Southern states only, white troops still had higher influenza and pneumonia morbidity than black troops (24.7% vs. 15.4%) (here we do not have the data to check for statistical significance). In the South, white soldiers also had much higher morbidity and death rates than white soldiers from other states, in particular compared with white soldiers from the Pacific Coast and the Rocky Mountain states, where the rates were the lowest in the country. Finally, these results showing that black soldiers from most states had higher morbidity from influenza and primary pneumonia are somewhat in contrast to the overall figures presented above, showing small racial differences in morbidity from influenza with and without complications (broncho- and lobar pneumonia, and others) in the US and even lower morbidity in Europe among black than white soldiers.

The last five studies analyzed data in the period in which the pandemic had the greatest impact, that is, in the fall of 1918. As calendar years may hide important epidemiological information, these studies are better at teasing out the real impact of the pandemic by race than that done by Howard and Love [8].

Britten [10] showed that for all soldiers in US army camps from September to December 1918, white people had 1.17 times higher incidence of total respiratory diseases (influenza, bronchitis, bronchopneumonia, lobar pneumonia) compared to black people (31.67% vs. 26.9%, a rate difference of 4.6, with 95% CI 4.07–5.14, z = 16.85).

Opie et al. [11] did a study of Camp Funston in Kansas with a special focus on three early waves of influenza in the spring of 1918. The first reports occurred in March in a white military unit, and then the disease spread to three other white military units. A new wave in April was rampant in both black and white soldiers, while in May, influenza spread only among new white recruits. From July to August, there was an outbreak of pneumonia, which may have been connected to influenza. The highest incidence from pneumonia occurred among black people (0.53% among black vs. 0.05% among white), new recruits, and especially black males from the states of Mississippi and Louisiana.

Brewer [9] studied Camp A.A. Humphreys, Virginia. In this camp, the white and black populations were divided into different organizations within the camp. This was normal practice in all military camps at the time. In Camp A.A. Humphreys, white troops lived in barracks and the black troops in tents. Brewer found that there were two cases of influenza among white troops in July and 15 more cases in August 1918, while just a few sporadic cases were reported among the black troops. Further, influenza morbidity in each of the five weeks during the main wave from September 13th to October 18th was higher among the white than among the black organizations: White soldiers had on average 2.3 times higher morbidity than black soldiers (17.7% vs. 7.6%, a rate difference of 10.1, with 95% CI 7.3–13.0, z = 6.99). There was a nonsignificant racial difference in the incidence of pneumonia (4.26% for white vs. 4.30% for black, z = 0.03). However, there were 2.39 times (95% CI 2.26–2.53, z = 34.21) more influenza cases complicated with pneumonia among the black troops than among the white troops (1.8 influenza cases per pneumonia cases vs. 4.2 influenza cases per pneumonia cases); this may therefore explain why the disease presumably was more fatal among the black cases than among the white cases [9].

Opie et al. [12] also studied Camp Pike, Arkansas, from 1 September to 31 October 1918, and found that influenza morbidity was 1.86 times higher for white than black troops (24.64% vs. 13.26%; rate difference of 11.38, 95% CI 9.26.13.50, z = 10.53), but there was a nonsignificant racial difference in the incidence of pneumonia (2.91% for white vs. 2.63% for black, z = 0.26). However, in Camp Pike, there were 3.42 times (95% CI 3.29–3.55, z = 51.76) more influenza cases complicated with pneumonia among the black troops than among the white troops (5.1 influenza cases per pneumonia cases vs. 8.5 influenza cases per pneumonia cases). In the study by Opie et al. [12], fatality from pneumonia by race was also estimated, but the difference was not significant (28.3% for blacks and 31.7% for whites, z = 0.53).

In Camp Pike [12], influenza morbidity was 30.58% among the new recruits (arriving after 20 August, most in September) and 15.50% among the veterans (in the camp before 20 August) (rate difference of 15.07, 95% CI 13.36–16.79, z = 17.20). This disparity in influenza morbidity was observed for both black soldiers and white soldiers. In one segregated black soldier tent colony at Fort Roots, within Camp Pike, influenza morbidity among recruits was 43.6%, about as high as any organization in Camp Pike, while veterans only had 7.6% morbidity, about equal to that of black soldiers of equal length of service in Camp Pike (rate difference between black recruit and black veterans was 35.94, 95% CI 29.71–42.16, z = 11.33).

Finally, Lucke et al. [13] studied Camp Zachary Taylor and Camp Knox, Kentucky, from September 22 to 1 December 1918, and found that morbidity from influenza was 3.29 times higher for white soldiers than for black soldiers (23.10% vs. 7.02%; rate difference of 16.07, 95% CI 13.31–18.83, z = 11.41), but there were no statistically significant racial differences in influenza mortality (0.72% for black vs 1.57% for white, z = 0.60) or influenza case fatality (10.23% for black vs. 6.78% for white, z = −0.67).

### 3.2. Insurance Populations

Frankel and Dublin [16] used insurance data to study monthly excess PI mortality by race, age (1–4 years, …, 70–74, and 75+) and gender October 1918 to June 1919. The calendar years 1911–1917 were used as the norm. The PI mortality rates in the pre-pandemic years 1911–1917 were higher for black people than for white people in every age- and gender-specific category. When studying the period from October 1918 throughout June 1919, however, crude observed mortality in ages 1–19 and excess mortality in ages 1–14 was higher for black people than for white people. In the same period, crude observed mortality for those aged 20–39 years and excess mortality for those 15–44 years was higher among the white people of both genders than among the black people of both genders. For persons older than 45 years, results for excess mortality by race were somewhat mixed, but crude observed mortality for those 40 years and older was in general higher among the black population. However, excess mortality for those 45 years and older declined by age for both races and both genders and approached zero and even negative values for those 70 years and older. In other words, excess mortality and the crossover in the role of race in mortality only occurred for those 20–40 years when the total pandemic period is considered.

When Frankel and Dublin [16] examined *monthly* data, another intriguing pattern emerged. They found that in October and November 1918, the crude observed mortality as well as excess mortality was still higher among the black population than the white population in the ages 1–14 years. For those aged 15–19 years, there were no racial differences in mortality. However, from age 20 and up to late 40s and early 50s, the mortality was higher among the white than among the black population. For those aged 25–29 years, for example, excess mortality for white males and white females was, respectively, 3.8% and 3.5%, and for black males and black females, respectively, 2.0% and 2.2%. Among these adult groups, we also see a higher mortality for white males compared to that for white females. However, gender differences were smaller among the black people in these same age groups. In each of the months from December 1918 to June 1919, the mortality rates turned back to normal with higher mortality for the black population vs. white population in each age and gender groups.

### 3.3. Populations in Representative Survey Data

One study analyzed both morbidity and case fatality for seven canvassed survey areas (Augusta, Minor Maryland communities, Little Rock, Baltimore, Spartanburg, Macon and Louisville). Results show that from September 1 until the end of 1918 or the beginning of 1919, the influenza morbidity rate after adjusting for age and gender was higher among the white respondent population than the black respondent population in all seven locations [17]. The lowest and highest influenza morbidity per locality for white respondents was, respectively, 17.9% and 45.6%, and for black respondents, it was, respectively, 4.9% and 38.5%. However, without controlling for age and gender due to few observations (deaths), case fatality rates in these seven communities were generally higher among the black people than among the white people. The lowest and highest influenza case fatalities per locality rate for white people was, respectively, 0.9% and 2.1% and for black people it was, respectively, 0.8% and 3.4%.

Another study considered influenza morbidity for eight canvassed areas by race and gender and pneumonia incidence by seven areas for race [10]. It included 101,132 respondents (79,766 or 78.9% white). In five of the eight areas (Baltimore, Louisville, Spartanburg, Macon, and Augusta), the influenza morbidity rates were higher among the white females than among the black females. However, in Charles County, minor Maryland towns/rural areas and Little Rock, the influenza morbidity rate was higher among black females than white females. For males, seven out of eight areas had higher morbidity for white people. The only exception is again Charles County, where black males had higher morbidity than white males. Data on cases and sample population by locality but not gender was available for significance testing. When doing the significance tests, we reproduced the findings reported above, and the racial differences were all statistically significant except for minor Maryland towns and Little Rock (tests not shown).

The pneumonia incidence in all of the seven surveyed populations (same communities as above except Charles county) was also higher among the white than among the black respondents, but none of the racial differences were statistically significant [10] (tests not shown). Finally, because both influenza incidence and pneumonia incidence tended to be higher for white people, the mortality rate (deaths per population in %) was slightly higher in white people than in black people. However, the influenza case fatality ratio (% of cases dying) was equal for white and black people (1.7% vs. 1.9%), while corresponding pneumonia case fatality was lower for white than for black people (28.8% vs. 39.8%) (no data to test for statistical significance).

### 3.4. Populations in Nationwide Routine Notification Data

We found three studies using mortality statistics from the national births and deaths registration areas in 1918. First, Weden [22] found that the 1918 influenza pandemic produced radical changes in the age distribution of survival in all subnational population groups. Weden also documented that 1918 is notably the only year in the twentieth century when differences between white people and black people in the age distribution of all-cause mortality were eliminated and life expectancy dropped precipitously. For all other years in the 20th century, black people had higher all-cause mortality than white people did.

Second, Garret [24] analyzed 14 US cities (Birmingham (Alabama), Atlanta (Georgia), Indianapolis (Indiana), Louisville (Kentucky), New Orleans (Louisiana), Baltimore (Maryland), Memphis and Nashville (Tennessee), Dallas and Houston (Texas), Norfolk and Richmond (West Virginia), Washington DC, and Kansas (Missouri)) and found higher influenza mortality rates for black people vs. white people in the pre-pandemic year of 1915 and also in the pandemic year of 1918 for all except Kansas in 1918. However, mortality in 1918 relative to 1915 was higher for white populations than for black populations. In other words, the rise in PI mortality in 1918 relative to baseline was higher for white than for the black populations in these 14 US cities.

Finally, Jordan [23] studied influenza mortality and pneumonia mortality in the calendar year of 1918 relative to the average of the seven calendar years 1911–1917 for both races. He found that mortality was higher for black people in 1911–1917 and in 1918 in the analyzed states of Kentucky, Maryland, North Carolina, South Carolina, and Virginia, but the racial differences were generally much smaller in 1918 than in 1911–1917. More interestingly, however, just like Garret [24], he documented that the rise in PI mortality in September–December 1918 relative to mortality in 1915 was clearly higher among the white people than among the black people in analyzed states such as Kentucky, Maryland, and Virginia, and in analyzed cities such as Birmingham, Atlanta, Louisville, Washington DC, New Orleans, Baltimore, Memphis, Nashville, and Richmond. For the three states and nine cities, lowest and highest rate ratios (1918 vs. 1915) for white people were, respectively, 9.8 and 18.3, and for black people, the corresponding figures were 4.6 and 11.1. The racial crossover in pandemic mortality in September–December 1918 from higher black to higher white mortality is most likely the reason why racial differences in mortality were smaller overall in 1918 than in the years 1911–1917, and also why Weden [22] found that differences between white people and black people in the age distribution of all-cause mortality were eliminated in 1918.

## 4. Discussion

### 4.1. What Did We Find?

First, for young male soldiers, black soldiers had lower pre-pandemic (1917) influenza morbidity than white soldiers (3% for black vs. 6% for white), but black soldiers had a higher incidence of lobar pneumonia than white soldiers (5% for black vs. 1% for white). During the 1918 pandemic, black soldiers had lower influenza morbidity both in the early and later waves (ca. 25% vs. ca. 13% in the fall of 1918)—as in the pre-pandemic situation—but black soldiers had (nonsignificantly) higher influenza mortality and case fatality than white soldiers. The black troops were, on the other hand, more susceptible to pneumonia than white troops, and a higher share of black influenza cases developed pneumonia. They had often, but not always, higher pneumonia morbidity, mortality, and case fatality both before and during the pandemic [7,8,9,10,11,12,13]. Humphreys [27] discussed whether the black population in the Southern states had more often been exposed to malaria, hookworm, and pellagra compared to the white population. When infected by one of these diseases, the chance of being more susceptible to bacterial diseases, like pneumonia, increased. In the Southern states, hookworm and malaria were common diseases in both white and black people, but it may have affected the black population to a higher degree. According to Humphreys [27], more of the black population lived in rural areas than in urban areas, and this “Rural Southerness” therefore made them less resistant to bacteriological diseases. This might therefore explain the tendency for a higher pneumonia incidence and higher share of influenza cases developing pneumonia in the black than in the white populations. As we discuss below, in the cities in the northern states, black people were more susceptible to influenza and pneumonia due to a higher prevalence of tuberculosis associated with severe household- and neighborhood-level crowding [28].

Second, one study of insurance holders and their families showed that PI mortality among white insurance holders was lower than for black counterparts at baseline (1911–1917) for both men (0.13% for white vs. 0.22% for Black) and women (0.16% for white vs. 0.17% for black). However, excess PI mortality in the last quarter of 1918 occurred only for those aged 20–40 years, and excess mortality was higher for the white population than among the black population in all of these ages. The highest excess mortality and the largest racial differences occurred for those aged 25–29 years (ca. 4% for white vs. ca 2% for black), while in 1919, mortality returned to the normal pre-pandemic pattern with higher mortality among the black population [16].

Third, survey data did not give baseline morbidity, but in the military, influenza morbidity was higher for white than for black in 1917 (6% vs. 3%). We may assume that this level of morbidity and the racial differences were equal among the civilian pre-pandemic populations. The surveys tapping morbidity in the fall of 1918 showed some variation by state, city, age, and gender, but in general, there was a higher morbidity for white populations (ca. 25% vs. ca. 13%), while case fatalities generally were higher for black populations (1–3% vs. 1–2%) [10,17]. This fatality was at least 10 times the fatality at baseline.

Finally, studies using data from the national death registration areas found that when calendar years were studied, racial differences in pandemic outcomes in 1918 were either small or almost eliminated [22], while Garret [24] and Jordan [23], studying the 1918 fall wave relative to 1915 data, documented clearly higher rise in the rates of mortality among the white population.

When we compare the results from all the different studies, we find that the white population had higher morbidity, but the black population had higher mortality and case fatality. The crossover in mortality is only visible when studying monthly mortality rates; PI mortality was higher for white people than for black people when we consider the peak pandemic period in the fall of 1918.

The number of deaths and mortality rates during influenza pandemics is a function of morbidity rates and case fatality ratios [29]. Morbidity is normally 5%–10% during historical and present-day seasonal influenza epidemics. In 1917, morbidity in the USA was at this level, but lower for black (3%) vs. white (6%) people. We do not know the level of case fatality in 1917, but presumably, it was higher for black people. This would explain why mortality rates in 1917 were higher for black people (0.22% for men, 0.17% for women) than for white people (0.13 for men, 0.16% for women). Morbidity during historical and recent pandemics, however, could be 40%–60%, or even higher among some sub-groups, and is the main reason for higher number of deaths and higher mortality during pandemics [29]. In the fall of 1918, morbidity in the US was lower for black than for white (ca. 13% vs. ca. 25%) people, but lethality was higher (1%–3% for black vs. 1%–2% for white). This combination of morbidity and case fatality would therefore explain why (excess) mortality for black was lower than for white groups (2% vs. 4% at ages 25–29 years). However, why morbidity was seemingly higher for black than for white people both at baseline in 1917 and before and during the spring/summer waves and later waves in 1918 is a question we address in the following section.

### 4.2. Explanation for Our Findings

Crosby [2,3] hypothesized that the black populations were less susceptible to the 1918 influenza pandemic autumn wave due to higher exposure to the less virulent spring/summer waves which left some immunity. In the following, we discuss how race may have been associated with disparities in, for example, household/community-level crowding and assortative mixing; and access to high-quality sanitation, water, hygiene, and handwashing. We also discuss how these factors subsequently may have been associated with differential exposure to the virus and morbidity in earlier waves and how potentially acquired immunity may have buffered against mortality in later waves.

#### 4.2.1. Direct Support for the Exposure Hypothesis

Even though a summer wave of influenza is not mentioned in previous reports on the surveys conducted in the United States [10,14,15], the survey data from Bergen, Norway, together with the recognition of higher morbidity in smaller towns compared to more urban areas in Maryland, support the idea that a pandemic wave before the autumn of 1918 may actually have hit the city of Baltimore as well [20]. This assumption is also supported by other studies showing that a first wave of influenza was present in the Northeastern United States in the spring/summer of 1918, for example, in New York City [30], and also in Camp Funston in Kansas [11]. Soldiers in Camp Funston exposed in March seemed less likely to be attacked in the subsequent April and May outbreaks. Black veterans and white veterans in Camp Pike, Arkansas, exposed to the summer wave gained some protection and had much lower morbidity in the fall wave than the newly arrived recruits [12]. Similar patterns of protection against the 1918 fall wave after being exposed to herald waves are described elsewhere [31,32]. The early cases in Kansas led some researchers to believe that the 1918 influenza pandemic first started in the USA and then spread to the rest of the world [2,3,33]. These observations in 1918 also are supported by more recent examples. Among military recruits in Singapore in 2009–2013, prior adenovirus or influenza virus infection conferred cross-protection against subsequent febrile illness episodes relative to prior infection due to other circulating viruses [34].

In order to substantiate the Crosby hypothesis that more black people were exposed to the mild summer wave than white people, we need empirical evidence on, for example: (1) Did more black people than white people live in cities, and especially, did more black people than white people live in and were they present in cities exposed to the spring/summer waves? (2) Did more black people than white people live in poor and cramped conditions within their homes and neighborhoods? (3) Did black people have poorer access to sanitation, water, and poorer hygiene and handwashing opportunities than white people? In other words, we need empirical information on these aspects where race may be a proxy for socioeconomic status.

In 1910, over 90 percent of the urban population in the United States consisted of white people [24]. In 1920, the majority of the Americans still lived in municipalities with no less than 2500 people [27]. Out of the white and black populations, 27% and 49% lived in urban areas, respectively. Most of the black population lived in the southern states in 1918. We have no information that a summer wave was especially rampant in the South. However, the Great Migration from rural South to northern cities, starting in 1916, meant that a larger share of black people lived in urban areas during the pandemic. Nevertheless, on average, fewer black people than white people lived in high population density urban areas where we can assume that more people were exposed to the spring/summer waves. Despite full segregation of black troops and white troops in US training camps, crowding in black organizations may actually have been less than among the white organizations because the black organizations had fewer individuals. In other words, one reason why influenza morbidity was lower among black troops may be due to the strong segregation.

The black people in the cities in the South often lived in dilapidated, crowded, and segregated housing areas. The Great Migration also left a large portion of the black people in cramped conditions in cities in northern states. One example is the city of Baltimore, where black migrant inner-city neighborhoods were struggling with deterioration and poverty and had less access to sanitation and urban social programs. The segregation created crowded urban ghettoes where tuberculosis became a prominent cause of death [28]. Individuals with such scarred lungs were likely more susceptible to both influenza and pneumonia and other infectious diseases [35,36,37]. This would also explain why a higher share of black influenza cases developed pneumonia and also the higher lethality from those diseases in 1918. When we take into account that a lower share of black people than white people lived in cities and that individuals of lower socioeconomic status may have poorer immune function due to more psychological stress, increasing their susceptibility of developing influenza given exposure [38], the relatively consistent result that black people had lower morbidity than white people is somewhat counterintuitive.

#### 4.2.2. Indirect Support to the Exposure Hypothesis

Crosby’s [2,3] hypothesis is that exposure to the disease agent early on primes the immune system to respond more effectively to the next insult. Several studies support such suppositions: Hörzer [39] documented that the first *victims* in the city of St. Paul, Minnesota, were reported in impoverished nonwhite areas, while later victims were of higher social status. This is also consistent with the finding for Oslo [40], where children from the affluent west side had higher morbidity during the major wave than during the early wave because they spent their summer holidays in the countryside where they were not exposed to the early wave and could therefore not acquire immunity to fight the second wave. The history for children on the less affluent east side of Oslo was the opposite. In Bergen, Norway, the first wave during the summer of 1918 hit the poor the hardest, while the rich with less exposure to the first milder wave was harder hit during the fall of 1918. At all socioeconomic status levels, males had the highest morbidity in the summer, while females had the highest morbidity in the fall [41]. Finally, in both Bergen and the two surveyed regions of Maryland, more females than males just above the age of 20 fell ill [20]. An explanation for this marked gender difference in both Bergen and Maryland could be that young male workers were more likely to be exposed to influenza during the first pandemic wave than females, who were mostly home-based, leading to males being better protected during following waves [20].

Excess mortality among US insurance holders from October to December 1918 was highest for young adults, and by race and gender, mortality was highest for white males, followed by white females, black females, and black males [16]. Placing these results for insurance holders in the context of the above research for Bergen and Maryland, it is likely that (1) a previously unrecognized pandemic summer wave may have hit the two regions of Maryland in 1918 but also other areas of the USA than New York and Kansas, and (2) that males of both races and black people were more exposed in the spring/summer of 1918, while females and white people with less exposure and acquired immunity during the spring/summer had higher morbidity and mortality in the fall of 1918. This would also be consistent with the assumption of a higher pre-pandemic exposure among black people, which would be one explanation for why white soldiers had higher influenza morbidity than black soldiers [7]. That positive excess mortality was mainly visible for those aged 20–40 of both races, that it peaked in those around the age of 30, and that excess mortality was negative for those 70+, is generally consistent with prior research [42]. Why mortality peaked at age 30 but declined into old age is still not clear. Explanations have included cytokine storms (multi organ failure), cardiac stroke and exposure early in life to other strains of influenza [43,44,45]. However, it is unclear why the highgly pathogenic virus should harm white young adults’ lungs and hearts harder than their black counterparts. It has been suggested that cohorts >30 years were protected because they were exposed to H1-like viruses (similar to the H1N1-virus in 1918) prior to the 1889 Russian influenza pandemic [42]. This prior exposure and subsequent protection may have been stronger for black people aged 30–40 than for their white counterparts, but as we have seen, both races had a decline in their positive excess mortality in their 50s and 60s, and negative excess mortality rates were observed for both races after 70+.

#### 4.2.3. Alternative Hypotheses

In our review, we did not find hypotheses other than the unequal exposure hypothesis suggested by Crosby [2,3] to explain the crossover in the role of race in 1918 pandemic outcomes. However, could this finding also be explained by concurrent events or secular changes in the quality of life and/or health care available to the black population? For example, is there any reason to believe that the explanation in the racial crossover in morbidity/mortality was due to any changes, in 1918–1919 or earlier, in how race was associated with disparities in psychological stress, nutritional status, and access to care providers/insurance, and by extension, differential susceptibility and disease severity? In other words, did WWI or other concurrent events lead to a relatively better position for black versus white people? One concurrent event was the Great Migration. However, as we have discussed above, the transition from rural to urban life made those of low socioeconomic status such as the majority of the black people more and not less susceptible to developing disease and to dying from it [35,38,46]. Further, is there any reason to believe that the quality of health care available to and treatment options used by the black and white patients changed in 1918, such that a racial crossover in mortality occurred due to differential consequences after the disease had developed? Because of generally poor living conditions in the South and troublesome conditions in northern cities due to the Great Migration, black nurses and black physicians were pushing for better sanitation laws and housing reforms to clean up the neighborhoods and the tenements [46]. In addition to this, “health weeks” were organized where black people could learn more about personal hygiene, and how they could avoid getting ill. However, a lack of resources and facilities prevented the efforts black nurses and other volunteers put into this program [46]. It was almost universal practice that the hospitals would either deny black people admission or they would be placed in a segregated space, like the basement or the attic. This forced the black people to develop their own health organizations, and by 1920, they managed to build several hospitals. Either way, the health care for black people was much poorer than for white people.

Another explanation of our findings might be selection. It can be assumed that the approximately 2 million young adult soldiers who were in France from September to December 1918 were the healthiest, leaving the frail behind [10,14,15]. By the spring of 1918, hundreds of thousands of troops were being deployed overseas monthly, and in November 1918, the draft and mobilization were at full capacity [47]. Among the medically examined, a larger share of black than white draftees were placed in Class I. Further, among the Class I members, a higher proportion of black individuals ended up in service. This unbalance was probably a consequence of discrimination, but on average, it might be reasonable to believe that this type of selection was stronger for the black than for the white population, leaving a frailer black civilian population behind. However, if this were true, it should have resulted in higher black PI mortality in the fall of 1918, not lower, as was observed. Moreover, the mortality of civilian males (and females) in urban and rural areas in the fall of 1918 in one of the canvassed areas for the surveys, Maryland, follows the same pattern and level of morbidity and mortality by gender as in Bergen in neutral Norway [20]. If a higher proportion of black than white draftees were sent to the front in Europe, this could potentially explain why morbidity and mortality was higher among the white civilian males in the fall of 1918. However, this was not the case, as half of both races were sent to France [5]. Finally, the selection hypothesis does not explain the higher white than black female pandemic outcomes in the fourth quarter of 1918, as nearly all soldiers were male.

One intriguing question is why mortality by race turned back to normal patterns exactly in December 1918, after only 2 months of deviation? Was this due to a deeper harvesting of white people than black people due to the higher white mortality in the fall of 1918, leaving more healthy white people at risk of dying during the winter and spring of 1919? The harvesting hypothesis gains some support from a study by Noymer and Garenne [36] showing that the high 1918 influenza mortality in the US took out those who were supposed to die from tuberculosis, lowering significantly the mortality from that disease in the 1920s. The return of soldiers in the months following the armistice in November 1918 may also explain the return to “normal” racial influenza outcome differences. However, troops from overseas did not return in large numbers until late spring and summer 1919, with the last division arriving in September 1919 [48]. Moreover, because they were only 3% of American combat forces, black soldiers suffered substantially fewer battlefield deaths and wounds than white soldiers [5]. Overall, black soldiers from the 92nd and 93rd combat divisions accounted for 773 of the 52,947 battlefield deaths sustained by the AEF in France during the war, less than 2% of all battlefield fatalities. Of American soldiers wounded, 4408 were black and 198,220 were white. White soldiers, therefore, made up nearly 98% of those wounded on the battlefield. The black soldiers probably suffered psychologically from the war as much as the white soldiers. However, the return of more physically wounded white than black soldiers should, everything else being equal, give the black group a mortality advantage; however, in 1919, mortality from PI and all causes were back to the pre-pandemic “normal”, with higher mortality for the black population.

A contemporary theory as to why the black population seemed to be less susceptible to upper respiratory tract diseases such as influenza and polio was that the lining of their noses (nasopharynx) was more resistant to microorganisms [7]. A more scientific and credible hypothesis, consistent with Crosby’s hypothesis, is that the black population on a general basis were exposed to flu viruses due to their social conditions and therefore had lower susceptibility during both the autumn and the spring/summer waves [46]. However, prior literature has missed that this also was the case before 1918, which would explain why white soldiers had higher influenza morbidity than black soldiers in 1910–17 [7]. When black people first got sick, they died more frequently, especially from bacterial complications such as pneumonia, likely because of their poor social conditions, seeking or receiving treatment at hospitals later, and a higher prevalence of concurrent illnesses (malaria and hookworm) compared to the white population [27].

A last alternative hypothesis is underreporting of influenza cases and deaths amongst the black population. It is impossible to determine how much of the lower influenza morbidity among the black people in survey data is due to the canvassing of areas with few black people and more complete reporting of white influenza cases [10]. However, the influenza cases in the surveys did not only include the severe cases brought to a doctor or a hospital. They rather included all cases the black homemaker could remember and report to the (white) enumerators. Thus, there is no reason to believe that morbidity (or mortality) data for black people are biased because cases were not brought to the attention of physicians due to discrimination by white doctors, access to few black doctors, or poor access to health care, health information, health insurance, and poverty. It is nevertheless worth noting that in Charles County, Maryland, where a little more than half of the population canvassed was black, morbidity was higher for both black males and black females compared to that for white males and white females.

One may assume that reporting of influenza cases is complete for both races in the military. However, in the highly segregated training camps in the USA, things were not as equal as believed by some in 1918 [7]. Black soldiers were usually placed in noncombatant labor battalions and received poorer housing, clothing, and food [5,6]. The army only allowed black officers to lead black troops, while white officers commanded both white and black units during the war. Out of 380,000 black soldiers, less than 1% or only 1353 served as officers, and only 1.5% of all military doctors were black [6]. Black people had poorer access to medical care and inferior care. Some black soldiers may have decided not to report themselves sick with influenza to a (possibly judgmental) white doctor, and independent black observers noted that white medical officers often delayed admitting black people into the hospital for treatment [5,6,27]. This may in sum explain lower hospital admissions for influenza but a higher rate of influenza cases developing into pneumonia in black soldiers [5]. The data in hospitals in France tell a contrasting story: Influenza morbidity in the calendar year of 1918 was actually *higher* for black troops than white troops [8]. In France, black soldiers were more likely to end up in labor battalions rather than in the trenches to fight. Although this was a consequence of ideas related to white supremacy [5,6], perhaps the stigma of black soldiers in the US training camps was less of an issue in France where interracial mingling was more accepted? Official surveys of soldier opinions of returning troops in 1919 support this idea. They show that black but not white soldiers expressed a preference for Europe over the United States in several ways. Black respondents, for example, did not complain about high prices in French stores. Instead, they focused on the fact that they were welcomed by every shopkeeper they encountered [5]. Finally, a number of diseases had higher pre-pandemic hospitalization rates for black than for white people, including lobar pneumonia and tuberculosis. Although these diseases were more severe than uncomplicated influenza, it shows that discrimination cannot explain racial differences in all causes of hospitalization.

Several major black states in the South were not covered in the National Births and Deaths Registration Area in 1918 [21]. Under-registration of black people is therefore an issue in the studies using official data for the US to study mortality (as in [22,23,24]). However, even in a major southern city such as New Orleans, white people had higher mortality than black people [23]. Finally, although insurance data may leave out more black people than white people, underreporting of deaths among black insurance holders is not likely to explain the crossover in mortality by race at end of 1918; why should underreporting only occur in the fall of 1918, and not in the pre-pandemic and post-pandemic phases, where PI mortality was higher for black people?

Crosby [2,3] suggested that black people due to poor and congested living conditions had higher prior exposure to the spring/summer wave in 1918. The higher morbidity at that time would therefore give the black people immunity, which would give lower morbidity and mortality in the fall. However, most of the black people lived in rural areas in the South, where there is no evidence for a herald wave in the spring of 1918. This would lead to little exposure to influenza before 1918 and to the herald waves in 1918 and thus less immunity and therefore higher morbidity and mortality in the fall of 1918. However, the observed data show the opposite. Another possibility is therefore that the lower black morbidity and mortality in the fall of 1918 occurred because Southern rurality led to a less widespread epidemic, protecting the black people who had low immunity. The black rural isolation may have led to less exposure, but when black people moved to northern cities and the men went off to war, they did get exposed. We can also make similar arguments taking the perspective of the white population. If a larger share was exposed in urban areas before 1918 and in areas that had herald waves, it should lead to higher immunity in white people. Why, then, was white morbidity higher than black morbidity in cities and in the military? Discrimination in the army is not likely to explain all racial differences in morbidity, and discrimination is less likely to be a factor in the lower reported black than white morbidity in surveys of the civilian populations.

Prior studies did not always control for possible confounders. Future studies on morbidity, mortality, and case fatality should therefore search in archives for the primary individual-level data for the military and the household-level data from the surveys undertaken in 1918. The primary level data from the Metropolitan Life Insurance Company may also exist in archives. Another possibility is to study linked census and death certificate data in order to study racial disparities in the time from illness onset to death or mortality during different pandemic waves, with controls for variables such as age, gender, marital status, occupational-based social class, housing standards or factors associated with disparities in exposure, susceptibility, and access to care. One possibility is to use already digitized data from a city such as Saint Paul [39], or Saint Louis, which had a substantial population of black people in 1918 (8%) [49]. Such studies would thus contribute to a better understanding as to why there was a crossover in the racial differences in pandemic outcomes that emerged between the third and fourth quarters of 1918, with a return to pre-pandemic greater excesses in black pandemic outcomes in December 1918.

## 5. Conclusions

The key reason for a higher number of deaths and high mortality rates during influenza pandemics is much higher morbidity than during seasonal influenza. Our review of studies using data on selected subgroups (soldiers and insurance holders) and from representative survey and routine notification data has shown that black people generally had lower PI morbidity and mortality rates but higher case fatality rates than white in the fourth quarter of 1918. However, pre- and post-pandemic influenza PI mortality rates were lower for white than for black people. The main reason for the crossover in the role of race in PI mortality rates was much higher morbidity for white than for black people. Despite proposed explanations related to social conditions and differential exposure, it is still unclear why these morbidity rates were higher. We also conclude that even though the black population had lower influenza incidence, they were more susceptible to other bacterial diseases, such as pneumonia. They therefore also tended to have higher case fatality rates than white people did.

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
