# Peer review of "Race and 1918 Influenza Pandemic in the United States: A Review of the Literature"

_ijerph, 2019, doi:10.3390/ijerph16142487_

Round 1
Reviewer 1 Report
This manuscript describes one host factor in regards to the 1918 inflluenza pandemic, that of race in the USA. The basic observation has been known for a century but the authors do a great service in pulling all the data together to be accessed despite their limitations. Their data collection and analyses match what I know about the 1918 pandemic and their conclusions are reasonable even if not all will agree with them.
My one overall criticism is that they emphasize social science aspects of race in America in 1918 which are likely correct but cannot be confirmed from the information presented. I refer here specifically to the "Great Migration" lines 470 and 551 and think that they should not base their conclusions on information which is only implied but not shown.
Another concluding point the authors could consider is that everything they present about racial differences involves the host and speaks against the hypothesis that a hyper virulent pathogen was the explanation of the unique mortality pattern seen during the 1918 pandemic.
Suggested corrections:
line 43 Would not claim first, try first to our knowledge etc
line 79 PI should be defined the first time it is shown
line 82 Change you to one
line 138 and many others: Need to be consistent about how one refers to USA towns and states. Suggest towns be identified with their respective state for those unfamilar with USA geography.
Line 240 Unclear to me why pneumonia does not show a racial difference but pneumonia following influenza does at Camp Pike. Vast majority of pneumonia would have been after influenza. Clarification would be good if the original paper allows it.
Line 353 The authors appear to have circular reasoning in terms of relating hookworm and malaria to susceiptibility to other infections. This needs to be clarified or deleted.
381 Suggest authors see recent JID paper from Singapore https://www.ncbi.nlm.nih.gov/pmc/articles/PMC6534195/ showing protection from subsequent febrile illnesses from previous adenovirus and influenza infections in military recruits. Likely this is analagous to what happened in 1918 and would give readers more used to having all the laboratory data some confidence that 1918 data has a basis in modern reality.
line 419 I don't think there is any evidence that access to medical care had any positive effect during the 1918 pandemic. On the contrary, those left in their barracks or cared for at home did better than those brought to the over crowded pneumonia wards. Although black people likely did have much less access to medical care than whites, I would not emphasize it as a likely explanation of differences in 1918.
Line 534 I realize it is traditional to call for more studies with more detailed data but since the 1918 pandemic was unique and predated our modern understanding of viruses, I am uncertain what this means in the context of this paper. If the authors know of other data sets or are working on analysing them, then they should say so.
Author Response
Dear Reviewer 1. Thanks for your extremely helpful comments. We are delighted to revise and resubmit our paper. We reply to your suggestions/comments in red text below.
All the best,
Authors
Comments and Suggestions for Authors
This manuscript describes one host factor in regards to the 1918 inflluenza pandemic, that of race in the USA. The basic observation has been known for a century but the authors do a great service in pulling all the data together to be accessed despite their limitations. Their data collection and analyses match what I know about the 1918 pandemic and their conclusions are reasonable even if not all will agree with them.
-Authors reply: Thanks so much.
My one overall criticism is that they emphasize social science aspects of race in America in 1918 which are likely correct but cannot be confirmed from the information presented. I refer here specifically to the "Great Migration" lines 470 and 551 and think that they should not base their conclusions on information which is only implied but not shown.
-Authors reply: We agree. We took this argument out of the conclusion.
Another concluding point the authors could consider is that everything they present about racial differences involves the host and speaks against the hypothesis that a hyper virulent pathogen was the explanation of the unique mortality pattern seen during the 1918 pandemic.
-Authors reply: Thanks for this comment. We decided not to bring this argument into the conclusion, as you suggest, “it is something we could consider”. We are not sure either, that studying race speaks against a hyper virulent pathogen as the key to understand the fall wave. Both races were exposed to this highly pathogenic virus in the fall of 1918, so the high mortality (experienced by both races) is of course partly due to this pathogen. However, it is not likely that black should react differently than the white people, independent of the context of WWI, prior immunity and concomitant diseases such as tuberculosis, when exposed to this pathogenic virus. The way we see it, the pathogenic virus in itself does not explain the racial differences in morbidity and mortality.
Suggested corrections:
line 43 Would not claim first, try first to our knowledge etc
-Authors reply: Done.
line 79 PI should be defined the first time it is shown
-Authors reply: Done.
line 82 Change you to one
-Authors reply: Done.
line 138 and many others: Need to be consistent about how one refers to USA towns and states. Suggest towns be identified with their respective state for those unfamilar with USA geography.
-Authors reply: We agree. Done.
Line 240 Unclear to me why pneumonia does not show a racial difference but pneumonia following influenza does at Camp Pike. Vast majority of pneumonia would have been after influenza. Clarification would be good if the original paper allows it.
-Authors reply: The data or text in the original paper does not give clarification on this. By the way, the same patterns were observed in Camp A.A. Humphreys, Virginia.
Line 353 The authors appear to have circular reasoning in terms of relating hookworm and malaria to susceiptibility to other infections. This needs to be clarified or deleted.
-Authors reply: We added the following sentence in the section “what did we find” to clarify our argument: “This might therefore explain the tendency for a higher pneumonia incidence and higher share of influenza cases developing pneumonia in the black than in the white populations”.
381 Suggest authors see recent JID paper from Singapore https://www.ncbi.nlm.nih.gov/pmc/articles/PMC6534195/ showing protection from subsequent febrile illnesses from previous adenovirus and influenza infections in military recruits. Likely this is analagous to what happened in 1918 and would give readers more used to having all the laboratory data some confidence that 1918 data has a basis in modern reality.
-Authors reply: Agree. Done.
line 419 I don't think there is any evidence that access to medical care had any positive effect during the 1918 pandemic. On the contrary, those left in their barracks or cared for at home did better than those brought to the over crowded pneumonia wards. Although black people likely did have much less access to medical care than whites, I would not emphasize it as a likely explanation of differences in 1918.
-Authors reply: We took out “less access to medical innovations”.
Line 534 I realize it is traditional to call for more studies with more detailed data but since the 1918 pandemic was unique and predated our modern understanding of viruses, I am uncertain what this means in the context of this paper. If the authors know of other data sets or are working on analysing them, then they should say so.
-Authors reply: We have explained that it is possible and important to search for the individual-level data from the insurance company, military and surveys in archives. We also show that it is possible and important to do individual-level studies using death certificate data linked with census data.
Reviewer 2 Report
This is a welcome review and analysis of the various studies of race and the 1918 flu in the US. It is a fascinating and important topic, that contributes to ongoing debates about race-related health disparities. The statistics from that era are, however, fraught with problems, and the authors might do more to include qualitative caveats in their discussion.
First of all, when compared to today’s race and ethnicity studies, there are no discussions about other racial/ethnic groups in the US—Native Americans, Hispanic Americans, or the large population of Asians already present on the west coast. As always, the authors are limited by the categories created and used by researchers ca. 1918. Still, it deserves a mention.
A second problem is the lumping together of blacks in northern urban areas and blacks in the rural south. Samuel Keller’s book Infectious Fear can give some background on the creation of so-called “lung blocks” in northern cities. Segregation created crowded urban ghettoes where TB became the prominent cause of death. Those with such scarred lungs were likely more susceptible to both influenza and pneumonia.
Blacks in southern rural areas were less crowded as most were rural. They were more prone to malnutrition, hookworm, pellagra and malaria, and may well have been anemic and otherwise compromised when they met the 1918 flu. Their rural isolation may have led to less exposure, but when their men went off to war, they did get exposed. This was also true for poor whites, however. One wants a category like southern sharecropper (white and black) rather than race. But this blending of socioeconomic, geographic, and racial characteristics is not going to found in contemporary studies. Still studies that gathered more northern urban blacks than southern ones are likely to be biased toward prior tb exposure.
Then we get to the biggest source of error of all. Who got counted? As the authors say, people in the military were probably the best counted. Rural poor folk of all sorts may have never seen a doctor in their life and if dead, were given their families’ assumption about diagnosis. One county in Georgia had a high rate of malaria deaths after world war II. When the fledgling CDC came in and tested supposed malaria patients, they all had other diseases. Malaria was a handy catchall.
Even today influenza is under-counted. Mild cases may never see a doc and choice to go to a doctor or a hospital will be affected by access to health care. Many mild pediatric cases in the US epidemic of H1N1 influenza of 2009 were never counted, although probably more did than in prior epidemics because anti-virals were available. So the mortality rate or hospitalization rate would underestimate the total population of the infected.
The discussion of influenza vs pneumonia among the military is similarly confused. If black troops did not get hospitalized until they were much further along in the disease than white troops, and if some believe, pneumonia is the final deadly event in most influenza, then such confusing stats would arise. They had no test for flu in 1918; they still debated whether it was bacterial or viral. Pneumonia on the other hand could be easily diagnosed from auscultation and sputum smears.
All of which is to say that the authors have done the best they can with the iffy data of the time. But a bit more flagging of potential confusions as noted above would add sophistication to their analysis.
Author Response
Dear Reviewer 2. Thanks for your extremely helpful comments. We are delighted to revise and resubmit our paper. We reply to your suggestions/comments in red text below.
All the best,
Authors
Comments and Suggestions for Authors
This is a welcome review and analysis of the various studies of race and the 1918 flu in the US. It is a fascinating and important topic, that contributes to ongoing debates about race-related health disparities. The statistics from that era are, however, fraught with problems, and the authors might do more to include qualitative caveats in their discussion.
-Authors reply: Thanks. We have added more on data quality, especially on the military. Here we refer you to our paragraph in data and methods (% of white vs. black medically examined, who were immediately qualified for service, % of those who eventually served, % of draftees by race who participated in the war etc), but also our discussion of lower hospitalization due to influenza morbidity, at least in the military, but not in the surveys, would be due to discrimination, few black doctors and lower expectations from the black in terms of quality of health care services offered.
First of all, when compared to today’s race and ethnicity studies, there are no discussions about other racial/ethnic groups in the US—Native Americans, Hispanic Americans, or the large population of Asians already present on the west coast. As always, the authors are limited by the categories created and used by researchers ca. 1918. Still, it deserves a mention.
-Authors reply: We agree. We therefore added this at the end of the paragraph describing the military data: “Finally, although there were 12,500 American Indians in the army and 20% of the draftees were foreign-born, including 5,700 Mexican Americans, with 25% being illiterate or not able to read English and with at least 9% non-US citizens [6], only “white” or “colored” (African American) were registered in the morbidity and mortality statistics and not ethnicity, citizenship, language spoken or literacy. As we will see in the next paragraphs, race is the major or only analytical category used in the insurance data, survey data and annual population data as well”.
A second problem is the lumping together of blacks in northern urban areas and blacks in the rural south. Samuel Keller’s book Infectious Fear can give some background on the creation of so-called “lung blocks” in northern cities. Segregation created crowded urban ghettoes where TB became the prominent cause of death. Those with such scarred lungs were likely more susceptible to both influenza and pneumonia.
-Authors reply: Thanks for this comment. We added a reference to Samuel Kelton Roberts book on tuberculosis in Baltimore.
Blacks in southern rural areas were less crowded as most were rural. They were more prone to malnutrition, hookworm, pellagra and malaria, and may well have been anemic and otherwise compromised when they met the 1918 flu. Their rural isolation may have led to less exposure, but when their men went off to war, they did get exposed. This was also true for poor whites, however. One wants a category like southern sharecropper (white and black) rather than race. But this blending of socioeconomic, geographic, and racial characteristics is not going to found in contemporary studies. Still studies that gathered more northern urban blacks than southern ones are likely to be biased toward prior tb exposure.
-Authors reply: Thanks for this comment. We have in the material and methods section, under the military data heading, explained that race often is the only analytical category that is available in the data. We write: Finally, although there were 12,500 American Indians in the army and 20% of the draftees were foreign-born, including 5,700 Mexican Americans, with 25% being illiterate or not able to read English and with at least 9% non-US citizens [6], only “white” or “colored” (African American) were registered in the morbidity and mortality statistics and not ethnicity, citizenship, language spoken or literacy. As we will see in the next paragraphs, race is the major or only analytical category used in the insurance data, survey data and annual population data as well”
We now also make it clear in the discussion that tuberculosis was a major risk factor for influenza among the black in northern cities, while malaria and hookworm were risk factors for influenza in the rural south.
Then we get to the biggest source of error of all. Who got counted? As the authors say, people in the military were probably the best counted. Rural poor folk of all sorts may have never seen a doctor in their life and if dead, were given their families’ assumption about diagnosis. One county in Georgia had a high rate of malaria deaths after world war II. When the fledgling CDC came in and tested supposed malaria patients, they all had other diseases. Malaria was a handy catchall.
-Authors reply: We agree on this. We have added on the possible under registration of influenza cases in the military due to discrimination and few black doctors. We also argue that
Even today influenza is under-counted. Mild cases may never see a doc and choice to go to a doctor or a hospital will be affected by access to health care. Many mild pediatric cases in the US epidemic of H1N1 influenza of 2009 were never counted, although probably more did than in prior epidemics because anti-virals were available. So the mortality rate or hospitalization rate would underestimate the total population of the infected.
-Authors reply: We agree on this. In the materials and methods section, under the heading “Survey data”, we therefore added this paragraph:
“All households were interviewed by trained staff. The influenza cases and deaths for a household were self-diagnosed and self-reported to the data collectors, usually by a homemaker, and was of course not lab-confirmed: in 1918 they did not know that influenza was caused by a virus, so even cases brought to hospital laboratories could not have been tested for whether it was caused by influenza or not. On the other hand, pneumonia cases could be easily diagnosed from auscultation and sputum smears. The influenza cases in the surveys did not only include the severe cases brought to a doctor or a hospital (or not brought to medical officers due to poor access to health care, health information, health insurance and poverty). However, just as would be the case even today, mild cases and asymptomatic cases were probably under-reported in these surveys for both races, as was also the case in official records from civilian/military doctors and hospitals”.
The discussion of influenza vs pneumonia among the military is similarly confused. If black troops did not get hospitalized until they were much further along in the disease than white troops, and if some believe, pneumonia is the final deadly event in most influenza, then such confusing stats would arise. They had no test for flu in 1918; they still debated whether it was bacterial or viral. Pneumonia on the other hand could be easily diagnosed from auscultation and sputum smears.
-Authors reply: We agree. See answer above, and the this added paragraph in the discussion:
“One may assume that reporting of influenza cases are complete for both races in the military. However, in the highly segregated training camps in the USA, things were not as equal as believed among some in 1918 [7]. Black were usually placed in noncombatant labor battalions, and received poorer housing, clothing and food [5,6]. The army only allowed African American officers to lead black troops, while white officers commanded both white and black units during the war. Out of 380,000 black soldiers, less than 1% or only 1,353 served as officers, and only 1.5% of all military doctors were black [6]. Black people had poorer access to medical care, inferior care, some may have decided not to report themselves sick with influenza to a (possibly judgmental) white doctor, and independent black observers noted that white medical officers often delayed admitting black people into the hospital for treatment [5,6,27]. This may in sum explain lower hospital admissions for influenza but a higher rate of influenza cases developing into pneumonia in black soldiers [5]. The data in hospitals in France tells a contrasting story: Here, influenza morbidity in the calendar year of 1918 was actually higher for black troops than white troops [8]. In France, black soldiers were more likely to end up in labor battalions rather than in the trenches to fight. Although this was a consequence of ideas related to white supremacy [5,6], perhaps the stigma of black soldiers in the US training camps were less of an issue in France where interracial mingling was accepted? Official surveys of soldier opinions of returning troops in 1919 supports this idea. They show that black but not white soldiers expressed a preference for Europe over the United States in several ways. African Americans, for example, did not complain about high prices in French stores. Instead, they focused on the fact that they were welcomed by every shopkeeper they encountered [5]”.
All of which is to say that the authors have done the best they can with the iffy data of the time. But a bit more flagging of potential confusions as noted above would add sophistication to their analysis.